# Efficacy of halopeRIdol to decrease the burden of Delirium In adult Critically ill patiEnts (EuRIDICE): study protocol for a prospective randomised multi-centre double-blind placebo-controlled clinical trial in the Netherlands

Lisa Smit [iD],[1] Zoran Trogrlić,[1] John W Devlin,[2,3] Robert-Jan Osse,[4] Huibert H Ponssen,[5] Arjen J C Slooter,[6] Nicole G M Hunfeld,[7] Wim J R Rietdijk [iD],[1] Diederik Gommers,[1] Mathieu van der Jagt,[1] on behalf of the EuRIDICE study group

For numbered affiliations see end of article.

**Correspondence to**
Dr Mathieu van der Jagt;
m.vanderjagt@erasmusmc.nl

## ABSTRACT

**Introduction** Delirium in critically ill adults is associated with prolonged hospital stay, increased mortality and greater cognitive and functional decline. Current practice guideline recommendations advocate the use of non-pharmacological strategies to reduce delirium. The routine use of scheduled haloperidol to treat delirium is not recommended given a lack of evidence regarding its ability to resolve delirium nor improve relevant short-term and longer-term outcomes. This study aims to evaluate the efficacy and safety of haloperidol for the treatment of delirium in adult critically ill patients to reduce days spent with coma or delirium.

**Methods and analysis** EuRIDICE is a prospective, multi-centre, randomised, double-blind, placebo-controlled trial. Study population consists of adult intensive care unit (ICU) patients without acute neurological injury who have delirium based on a positive Intensive Care Delirium Screening Checklist (ICDSC) or Confusion Assessment Method for the ICU (CAM-ICU) assessment. Intervention is intravenous haloperidol 2.5 mg (or matching placebo) every 8 hours, titrated daily based on ICDSC or CAM-ICU positivity to a maximum of 5 mg every 8 hours, until delirium resolution or ICU discharge. Main study endpoint is delirium and coma-free days (DCFD) up to 14 days after randomisation. Secondary endpoints include (1) 28-day and 1-year mortality, (2) cognitive and functional performance at 3 and 12 months, (3) patient and family delirium and ICU experience, (4) psychological sequelae during and after ICU stay, (4) safety concerns associated with haloperidol use and (5) cost-effectiveness. Differences in DCFDs between haloperidol and placebo group will be analysed using Poisson regression analysis. Study recruitment started in February 2018 and continues.

**Ethics and dissemination** The study has been approved by the Medical Ethics Committee of the Erasmus University Medical Centre Rotterdam (MEC2017-511) and by the Institutional Review Boards of the participating sites. Its results will be disseminated via peer-reviewed publication and conference presentations.

---

### Strengths and limitations of this study

► This study is the first sufficiently powered randomised, multi-centre, double-blind, placebo-controlled clinical trial in Europe.
► Extensive neurocognitive testing will be conducted with a valid test battery in order to assess cognitive impairment at 3 and 12 months after intensive care unit (ICU) admission.
► We will assess patient and family experiences associated with delirium as a novel outcome.
► There are little data on the optimal haloperidol regimen in ICU patients; the maximum haloperidol dose of 15 mg/day in our study may still be subtherapeutic.
► Lack of true clinical equipoise among nurses and physicians regarding the use of haloperidol may hamper motivation for the study.

---

**Trial registration** NCT03628391

## INTRODUCTION

Delirium occurs in up to 80% of patients admitted to the intensive care unit (ICU)[1 2] and is associated with greater ICU and post-ICU mortality.[2] Cognitive dysfunction and functional decline after critical illness is common, frequently persists for months after ICU discharge and is worse among patients who experience delirium.[2 3] The symptoms and sequelae of delirium, including fear, anxiety, disrupted sleep and post-traumatic stress disorder, may persist for months after ICU discharge. The health and societal costs of delirium are estimated to exceed $10 billion per year in the USA alone.[4]

Given the burden and costs of delirium in critically ill adults, substantial research efforts have been devoted to identify safe and effective strategies to treat it. Current evidence and practice guideline recommendations advocate the use of non-pharmacological strategies to reduce delirium, including avoidance of benzodiazepine sedation, early mobilisation and the use of sleep improvement protocols. The routine use of medication-based interventions to treat delirium, other than treatments to reduce the agitation that sometimes accompanies it, are not recommended.[5 6] The routine use of scheduled haloperidol to treat delirium is not currently recommended given a lack of current evidence regarding its ability to resolve delirium and its symptoms, nor improve relevant short and longer-term outcomes.

At the time this protocol was finalised, two randomised, placebo-controlled trials had evaluated haloperidol for ICU delirium prophylaxis or treatment and found haloperidol use did not affect days spent with delirium, days of mechanical ventilation, nor time spent in the ICU or hospital.[7 8] In one of these randomised controlled trials, haloperidol use was associated with less agitation.[7] Importantly, both studies were small (a combined total of 212 patients were enrolled), the ABCDEF bundle (a multimodal ICU bundle shown to reduce delirium by 50%)[9] was not routinely used, the effect of haloperidol on delirium-related symptoms was not evaluated and the post-ICU, longer-term outcomes were not considered. Whether the response to haloperidol was different between patients with hyperactive versus hypoactive delirium was also not evaluated. The impact of haloperidol on patients' and families' experiences with delirium after ICU discharge remains unknown. Whether long-term mortality is causally related to delirium or simply the persistent cognitive and functional decline associated with critical illness can only be established through a randomised trial.[10] Moreover, the use of haloperidol in critically ill adults is not without potential safety concerns given it may prolong the QTc interval, induce extrapyramidal effects and cause oversedation. Despite haloperidol's lack of proven efficacy and the safety concerns associated with its use, haloperidol continues to be widely used in ICUs to treat of delirium.[11]

In light of the aforementioned evidence gaps that were identified at the time this trial was conceptualised, there is a clear need for a large, multi-centre, randomised controlled trial to better define the efficacy and safety of haloperidol to treat delirium in critically ill adults. This report describes the protocol for a large, multi-centre, randomised, placebo-controlled, haloperidol delirium trial that recently started enrolling patients across multiple ICUs in the Netherlands.

## METHODS AND ANALYSIS
### Study design
Randomised, double-blind, placebo-controlled trial of haloperidol for the treatment of delirium in patients admitted to one of six participating ICUs in the Rotterdam area in the Netherlands. See online supplemental appendix 1 for the participating hospitals.

### Study population
Consecutive adults admitted to one of the participating ICUs.

### Eligibility criteria
#### Inclusion criteria for eligibility
1. Age ≥18 years
2. Admitted to the ICU.

#### Exclusion criteria for eligibility
1. Admitted to the ICU with an acute neurological diagnosis (including acute stroke, traumatic brain injury, intracranial malignancy, anoxic coma). Prior non-acute stroke or another neurological condition without cognitive deterioration is not an exclusion criterion.
2. Pregnancy or lactation.
3. History of ventricular arrhythmia including 'torsade de pointes' (TdP).
4. Known allergy to haloperidol.
5. History of dementia or an Informant Questionnaire on Cognitive Decline in the Elderly (IQCODE) score ≥4[12]
6. History of malignant neuroleptic syndrome or parkinsonism (either Parkinson's disease or another hypokinetic rigid syndrome).
7. Schizophrenia or other psychotic disorder.
8. Inability to conduct valid delirium screening assessment (eg, coma, deaf, blind) or inability to speak the Dutch language.
9. Expected to die within 24 hours or leave the ICU within 24 hours.

#### Inclusion criteria for randomisation
1. Delirium, as assessed with the Intensive Care Delirium Screening Checklist (ICDSC ≥4) or the Confusion Assessment Method for the ICU (positive CAM-ICU assessment), at the time of ICU admission or any ICU day after ICU admission.
2. Written informed consent obtained from the patient or their legal representative.
3. All eligibility *in*clusion criteria (from above) are still met.

#### Exclusion criteria for randomisation
1. Prolonged QT-interval (QTc >500 ms).
2. (recent) TdP.
3. (recent) Neuroleptic malignant syndrome or parkinsonism.
4. Evidence of acute alcohol (or substance) withdrawal requiring pharmacological intervention (eg, benzodiazepines or alpha-2 agonist) to treat.
5. The patient is expected to die within 24 hours or expected to leave the ICU within 24 hours.
6. No (previously) signed informed consent by patient or representative.

7. Current participation in another intervention trial that is evaluating a medication, device or behavioural intervention.

## Study outcomes
### Main study outcome
ICU delirium and coma-free days (DCFDs) (up to 14 days after randomisation).

### Secondary study outcomes
During ICU stay
► Richmond Agitation Sedation Scale (RASS).
► Maximum ICU Mobility Scale (IMS[13]) and day of maximum IMS.
► Quality of sleep (Richards-Campbell Sleep Questionnaire (RCSQ)[14] and with a visual analogue scale between 1 and 7 assessing the sleep quality according to the nurse).
► Use of 'escape medication' for hallucinations and/or agitation (including atypical antipsychotics, alpha-2 agonists, GABA agonists, opiates and 'open-label' haloperidol).
► Daily study drug dose corrected for body weight (mg/kg).
► Self-extubation rate, removal of invasive devices (intravenous/intra-arterial catheters, drains and tubes).
► Adverse drug-associated events (prolonged QTc by EKG, muscle rigidity and other associated movement disorders (Simpson Angus Scale[15]) and ventricular arrhythmias including TdP).
► Blood pressure will be recorded previous to and 1 hour after the first study drug dose (2.5 mg equivalent) and 1 hour after the first 5 mg equivalent.
► Daily respiratory status (regarding endotracheal intubation and mechanical ventilation).
► Time from randomisation to first resolution of delirium.
► Time to 'readiness for discharge from the ICU'.
  Hospital discharge
► Patient and family member well-being and experiences associated with delirium during and after ICU stay with the ICU Memory Tool (ICU-MT[16]) and Delirium Experience Questionnaire (DEQ[17]).
  28 days after randomisation
► Mortality rate.
  3 months after randomisation
► Cognitive outcomes with a detailed cognitive assessment battery of validated and repeatable measures of general cognition, memory, language, processing speed, attention and executive functioning, and mood (Montreal Cognitive Assessment (MOCA),[18] Rey Auditory Verbal Learning Test,[19] Semantic fluency,[20] Digit Span (WAIS-IV),[21] Trail making tests A and B,[22] Boston naming Test (short version),[23] Hospital Anxiety and Depression Scale (HADS)[24]).
► Functional outcomes and quality of life (Short Form-36 (SF-36)[25]).

► Patient and family member well-being and experiences associated with delirium during and after ICU stay with the ICU-MT,[16] DEQ[17] and Caregiver Strain Index (CSI[26]).
► Post-traumatic stress syndrome in participants and family members with the Impact of Event Scale–Revised.[27]
  12 months after randomisation
► Cognitive outcomes with a detailed cognitive assessment battery of validated and repeatable measures of general cognition, memory, language, processing speed, attention and executive functioning, and mood (MOCA,[18] Rey Auditory Verbal Learning Test,[19] Semantic fluency,[20] Digit Span (WAIS-IV),[21] Trail making tests A and B,[22] Boston naming Test (short version),[23] HADS[24]).
► Functional outcomes and quality of life (SF-36[25]).
► Mortality rate.

A cost-effectiveness analysis will be performed in collaboration with the Department of Health Policy and Management of Erasmus University Rotterdam (see online supplemental appendix 2 for more detailed explanation). The tools for the secondary outcomes are mentioned in table 1 with overview of timing of assessments.

## Treatment of subjects
### Investigational product
Name: Haldol (haloperidol)

Mechanism: butyrophenone-derived anti-psychotic with mainly dopamine-2 receptor antagonistic properties

Placebo consists of sodium chloride for injection. Medical staff, patients and family will be blinded to the product containing haloperidol/placebo.

### Summary of findings from clinical studies and of known and potential risks and benefits
See: Summary of Product Characteristics in online supplemental appendix 3 and Systematic Review (online supplemental appendix 4).

### Dosages, dosage modifications and method of administration
The following dosing scheme will be used: start with haloperidol/placebo (further called: 'study drug') 2.5 mg intravenously q8h (because of delirium screening once every 8-hour shift) and increase to a maximum dose of 5 mg intravenously q8h when delirium persists during the next 8-hour shift. Doses will be reduced (50% of dose) in the very old elderly (age ≥80 years). The study drug dose will be decreased (when dosage is 5 mg intravenously q8h) or stopped (when dose is 2.5 mg intravenously q8h) when delirium has resolved (or is un-assessable due to coma) for the next 24 hours (implying: three consecutive delirium assessments during three shifts). Dosages can be lowered also at the discretion of the treating physician in case of evident rigidity, which is in line with current routine practice. Standard clinical practice for the administration of haloperidol will be followed.

**Table 1** Overview of timing of assessments, including required time investment per visit/questionnaire

| Moment (months) | Neurocognitive tests | Patient and family experiences (time in min) | Functional outcomes (SF-36) | Cost-effectivity questionnaires (EQ-5D-5L, iMTA MCQ, iMTA PCQ) | Other |
|---|---|---|---|---|---|
| Enrolment | | | | | Informed consent, IQCODE-N, pregnancy test (if applicable), EKG |
| ICU study period (3×/day) | | | | | CAM-ICU/ICDSC, RASS |
| ICU study period (once daily) | | | | | IMS, RCSQ. Only when on study medication: EKG, Simpson Angus Scale |
| 0 (discharge from hospital) | | Patient: ICU-MT[15]+DEQ[15] Family: DEQ[2] | | | |
| 1 | | | | 30 min | |
| 3 | 45–60 min | Patient: IES-R[5]+ICU MT[15]+DEQ[15] Family: IES-R[5]+CSI[5]+DEQ[2] | 10 min | 30 min | |
| 6 | | | | 30 min | |
| 12 | 45–60 min | | 10 min | 30 min | |

Neurocognitive tests: Montreal Cognitive Assessment (MOCA), Rey Auditory Verbal Learning Test, Semantic fluency, Digit Span (WAIS-IV), Trailmaking tests A and B, Boston naming Test (short version), Hospital Anxiety and Depression Scale (HADS).
Simpson Angus Scale=measures muscle rigidity and other associated movement disorders.
With the exception of the neurocognitive tests, all aforementioned tools are questionnaires that can be administered at home. Real-life visits only need to be paid in order to perform the neurocognitive tests.
CSI, Caregiver Strain Index, assesses the strain experienced by the caregiver; DEQ, Delirium Experience Questionnaire, measures experiences linked to delirium; EKG, Electrocardiography; EQ-5D-5L, assesses the general health status; ICU-MT, ICU-Memory Tool, assesses the experience and memories of ICU admission; IES-R, Impact of Event Scale Revised, assesses distress linked to a traumatic experience (i.e. experiencing delirium); IMS, ICU Mobility Scale, measures mobility during ICU admission; iMTA MCQ, instituut Beleid & Management Gezondheidszorg Medical Consumption Questionnaire (healthcare use); IQCODE-N, Informant Questionnaire on Cognitive Decline in the Elderly – Dutch version; RCSQ, Richards-Campbell Sleep Questionnaire, measures quality of sleep; SF-36, Short Form-36, measures the health-related quality of life.

### Description and justification of route of administration and dosage

Administration of the study intervention via the intravenous (vs the oral or enteral) route is the most feasible in critically ill patients—a population where gastrointestinal dysfunction is prevalent and haloperidol absorption (ie, bioavailability) could be compromised. The dose of haloperidol or placebo equivalent to be used in the study is based on the following consideration: (1) Pharmacokinetics/pharmacodynamics; (2) efficacy; (3) safety. A (pilot) study in Erasmus Medical Centre (n=14 critically ill patients, abstract presented at European Society of Intensive Care Medicine 2016) showed no adverse events (eg, no QTc >500 ms), low serum levels (1.5–2.2 µg/L) and no clear relation between serum level and delirium resolution with haloperidol dosages up to 2 mg intravenously q8h (or: 3×2 mg intravenously). A feasibility trial of haloperidol for ICU delirium (MIND trial[8]) that used an average total daily dosage of 15 mg orally found higher serum levels (IQR 2.85–5.8 µg/L). No differences were found in QTc prolongation between treatment groups and placebo in this trial. None of these trials demonstrated clinically important safety concerns associated with haloperidol administration. Finally, a recently published trial of haloperidol for ICU delirium using haloperidol/placebo 10 mg intravenously q12h did not report any safety issues, using a QTc cut-off for safety of 550 ms, which may be regarded an indirect signal that such dosages are feasible and safe.[28] The maximum dose of haloperidol of up to 5 mg intravenously q8h was further chosen because a previous Dutch guideline advocating the use of haloperidol recommended an intravenous haloperidol treatment dose of up to 20 mg/24-hour period.[29] In our protocol, we chose q8h dosing (titrated up to 15 mg daily) given the greater potential susceptibility of critically ill adults to the side effects of haloperidol and the fact that this dosage is in line with existing haloperidol delirium protocols in several of the participating ICUs.

## Patient assessments

Rigidity will be monitored with the Simpson-Angus scale[15] and the Barnes Akathisia Rating Scale[30] (see 'Secondary study endpoints' section) for study purposes only. The QTc interval will be measured daily before the administration of the second daily (afternoon) dose using a 12-lead EKG. When the QTc interval is found to be prolonged (>500 ms or an increase from baseline (=at randomisation) of ≥60 ms[31 32]), all non-study medications having the potential to prolong the QTc will be held if clinically feasible. A Standard Operating Procedure (SOP) lists the drugs known to prolong the QTc. Eight hours later, if QTc prolongation persists, study medication will be held or tapered according to the SOP and only resumed when the EKGs (evaluation frequency increased to q8h in this situation) reveal QTc prolongation to have dissipated.

## General medical management at participating ICUs

In the six original participating ICUs, institutional delirium guidelines, based on the 2013 PAD guidelines and a Dutch ICU delirium guideline, were rigorously implemented over a 3-year period (2012 to 2015).[6 33 34] During the inclusion period of the current trial, spot-checks will be performed by members of the investigative team at each centre to confirm delirium screening accuracy as a quality-of-assessments measure, and these will be documented in a qualitative manner.

## Preparation and labelling of investigational medicinal product

Preparation and labelling will be done by the trial pharmacist ('Apotheek A15') according to Good Manufacturing Practices guidelines. Apotheek A15 is certified for these procedures. Trial medication will be dispensed to the pharmacies of the trial sites by the Hospital Pharmacy of Erasmus MC. See online supplemental appendix 5 for a description of the drug accountability.

## Escape medication

Knowing that half the subjects will be administered placebo, we anticipate two issues may affect the clinical management of enrolled patients: (1) agitation and (2) hallucinations.

Agitation management will be based on the following principles: (1) treat pain first with opioids; (2) use alpha-2 agonist for agitation that either persists or is not caused by pain; (3) GABA agonists (eg, benzodiazepines or propofol) are discouraged, but can be used on a short-term basis for the treatment of severe agitation (RASS ≥2) that cannot be effectively managed by other means.

Hallucination management will be based on the following principles: (1) pharmacological treatment may be withheld if the patient indicates they are not in distress; (2) for a patient in distress, a low-dose atypical antipsychotic (eg, quetiapine 12.5 mg q8h) may be administered on a short-term basis until the distress resolves.

Because of the pragmatic design of this trial, within these boundaries, the treatment and dose of escape medication is left to the treating physician since these are part of routine practice. However, before start of randomisation, these management principles for agitation and hallucination will be thoroughly implemented first with the help of detailed SOP's to enhance uniformity in participating centres. Adherence to escape medication regimens will be closely monitored. Open-label haloperidol administration is strongly discouraged during the trial but can be used if the ICU team considers it necessary for acute breakthrough delirium symptoms that cannot be managed within the management boundaries outlined previously. Open-label haloperidol will be documented.

## Randomisation, blinding and treatment allocation

Legal representatives of eligible patients (when the patient is sedated or otherwise temporarily unable to consent) or the patient himself/herself will be asked for informed consent shortly after admission when the patient has no delirium or as soon as possible after admission when the patient already has delirium. Online supplemental appendix 6 contains an example of the patient consent form. In this study, the presence of delirium will be considered to be confirmed when the bedside nurse deemed the patient to have delirium based on assessment with the ICDSC or CAM-ICU, given the previous large-scale implementation project.[33]

Delirious patients who fulfil all inclusion but no exclusion criteria, and for whom written informed consent has been obtained (as recorded in medical file), will be randomised. Randomisation coordination and start of a new case record form (CRF) will be guided by the electronic data capture (EDC) system of ALEA, constructed by the Clinical Trial Centre of the Erasmus Medical Centre and calibrated with the coordinating (Erasmus MC) and local pharmacies. We will randomise the recruited patients using a block design of eight patients in one block, and one block is assigned to a centre. We will have eight batches (numbered 1 through 8) of treatment and placebo, with four batches of placebo and four treatment (haloperidol). Each block will have a random assignment of eight batch numbers, having four placebo and four haloperidol patients included (a combination of 1 to 4 and 5 to 8 in random order). After eight patients are included in the study (ie, a block is full), a new block will be assigned to a centre.

On randomisation, the study drug with the corresponding randomisation kit numbers 1–8 (based on eight medication batches consisting of either haloperidol or placebo) will be obtained from the hospital pharmacy of each participating ICU. Each box from a batch/kit contains 10 ampules (5 mg/1 mL) of haloperidol or placebo. If all ampules are used, a new box from the same medication kit number with 10 ampules will be used. Study drugs are administered on prescription in the electronic patient data management system (PDMS) and are double-checked by ICU nurses before administration, which is similar to regular practice. Furthermore, the kit number was noted on randomisation in the medical file

and the kit number could be retrieved at any time from the PDMS after first prescription on randomisation.

Blinding of the medication will be performed by the pharmacy, based on a randomisation list that will be generated electronically through a randomisation module in the EDC system of ALEA. Randomisation will be stratified per study centre (ie, equal number of patients in both study groups; see 'Statistical analysis' section). Only the involved pharmacists and the trial statistician are aware of the contents of each medication kit. Only the local (site) pharmacists are able to unblind study treatment of a patient in case of an emergency. Except for the hospital's pharmacist responsible for the randomisation list, all other involved personnel with the study, caregivers, patients or their representatives will remain unaware of the treatment groups until the time of Database Lock. The unblinding procedure is specified in online supplemental appendix 7.

Follow-up procedures will be performed according to designated SOPs. When possible and preferred by patients or families, questionnaires will be sent or visits planned at home when possible, for example, for incapacitated participants.

### Withdrawal of individual subjects

Subjects can leave the study at any time for any reason if they wish to do so without any consequences. The investigator can decide to withdraw a subject from the study for urgent medical reasons.

### Follow-up of subjects withdrawn from treatment

Data of withdrawn patients will remain in the database for statistical analysis purposes but will not be subject to follow-up. When patients specifically withdraw their consent for usage of their data, these data will be removed from the database and excluded from all analyses.

### Premature termination of the study

The sponsor may decide to terminate the study prematurely based on the following criteria:

► There is evidence of an unacceptable risk for study patients (ie, safety issue).
► There is reason to conclude that continuation of the study cannot serve a scientific purpose following confirmation of the Data Safety Monitoring Board (DSMB).
► The DSMB recommends to end the trial based on viable arguments other than described above.

The following stopping rules have been determined by the DSMB and have been laid down in a DSMB charter:

► Early stopping of one individual participant, for example, to clear benefit or harm of a treatment or the occurrence of serious adverse reactions or events in one patient. In this case, de-blinding of this single patient may be necessary.
► Stopping of the trial as a whole to clear benefit or harm of a treatment or the occurrence of serious adverse reactions or events. As a result, further patient

enrolment will be stopped. De-blinding may be necessary for all patients.

Reasons to stop the study include

► Advice to do so from DSMB.
► Interim analysis shows a significant benefit difference between the treatment groups which will not be expected to change after inclusion of all subject as per the power analysis.

Suspected unexpected serious adverse reactions (SUSARs) are not expected due to the vast experience in clinical practice with the study drug (haloperidol).

If the study is terminated, the Medical Ethics Committees of all participating hospitals and the Central Committee on Research Involving Human Subjects (CCMO) will be notified.

### Safety reporting
#### Adverse events (AEs), serious adverse events (SAEs) and SUSARs
*Adverse events*

AEs are defined as any undesirable experience occurring to a subject during the study, whether or not considered related to the investigational product. Since patients admitted to an ICU are critically ill and present with many AEs, only possible adverse drug-related events (on days of study drug administration: prolonged QTc by EKG, muscle rigidity and associated movements disorders (Simpson Angus Scale)), as indicated by the subject or observed by the investigator or his staff occurring from the date of randomisation until 14 days later or discharge from ICU or death (whichever comes first), will be recorded in the CRF. In addition, the following AEs will be assessed daily during 14 days after randomisation: epilepsy, tachycardia, hypotension (not explained otherwise), hepatic dysfunction (not explained otherwise), leucopenia (not explained otherwise) and bronchospasms (not explained otherwise).

#### *Serious adverse events*

An SAE is any untoward medical occurrence or effect, occurring during the 14-day study period at the ICU, that (the SAEs for the purpose of the study are shown in *italics* per item)

► Results in death;
 – Death will always be reported as an SAE
► Is life threatening (at the time of the event);
 – *Ventricular arrhythmia or malignant neuroleptic syndrome*
► Requires hospitalisation or prolongation of existing inpatients' hospitalisation;
 – *Not to be expected; only applicable when the site investigator is able to explicitly show a relationship*
► Results in persistent or significant disability or incapacity;
 – *Not to be expected; only applicable when the site investigator is able to explicitly show a relationship*
► Is a congenital anomaly or birth defect; (*Not applicable*) or

► Any other important medical event that did not result in any of the outcomes listed above due to medical or surgical intervention but could have been based on appropriate judgement by the investigator.

## Statistical analysis

### Primary and secondary study parameter(s)

Statistical analysis will be done according to intention-to-treat principle. All randomised participants will be included. The primary outcome is DCFDs, defined as the number of days in the first 14 days after randomisation during which the patient is alive without delirium and not in coma from any cause.[7] Patients who are discharged before the 14-day study period has ended will be recorded as delirium and coma free after discharge.[8 35] In addition, we will assume all patients who died within 14 days after randomisation to have 0 DCFDs.[7] Differences between DCFDs between the haloperidol group and placebo group will be analysed using Poisson regression analysis, with adjustment for differences in baseline characteristics between treatment groups (when present) and for the different centres. We will collect data with regards to baseline demographics: age, sex, admission diagnosis category, APACHE II and APACHE IV, Sequential Organ Failure Assessment (SOFA) score, ICU days before study entry and pre-admission delirium duration in participants with delirium on admission. Pre-defined sub-analyses will include efficacy stratified by (1) agitated, mixed-type or hypoactive delirium; (2) the presence of hallucinations or delusions; (3) delirium severity (based on ICDSC score: low delirium severity=mean ICDSC score of 4 to 5; medium delirium severity=mean ICDSC score 5 to 7; high delirium severity=ICDSC score 7 to 8); and (4) sedation-related, hypoxic, metabolic or septic delirium. For cognitive and functional outcomes assessed with designated test batteries, non-parametric or parametric tests will be used depending on normality of scaled test results. Mortality risk will be assessed as a binary endpoint. A more detailed statistical analysis plan, to be drawn up before Data Base Lock, will be drafted for publication separately.

## INTERIM ANALYSIS

Pre-planned interim analyses will be performed at one-third and two-thirds of the trial's course (first analysis ideally estimated at 6 months after start of trial), as determined by the DSMB charter or otherwise when the DSMB requests it.

## Sample size calculation

To achieve statistically significant results (p<0.05) with a power of 90% and a true treatment difference of 1 day for the primary outcome (from 3.2 DCFDs in the placebo group to 4.2 in the haloperidol group, SD in both groups is equal to 4.2), 371 patients are needed in each group (n=742). These estimates are derived from the previous implementation study, which included 4727 patients in

three 4-month periods in the same six participating ICUs and found delirium incidence of 27% (and increase of DCFDs from 60% to 70%).[33] Consequently, presuming an informed consent rate of 40%, we need 18 months to encounter 1900 patients with a newly diagnosed delirium to include the required 742 patients. Because of estimated workload due to follow-up visits, including, for example, neurocognitive testing, we propose to select a convenience sample of two-thirds of ICU survivors (estimated around 400 of 575 survivors) as a random sample for the cognitive, functional and secondary outcome variables.

## Patient and public involvement

During the design and conduct of the study, we involved two ex-ICU patients as patient-perspective representatives. The primary research question, its outcome measures and the burden of the intervention have been assessed and found relevant by these patient-representatives. The role and tasks of the patient-representatives for the study have been detailed as (1) to help select meaningful assessment tools of patient and family experiences during and after ICU stay; (2) act as liaison between the study management team and the Dutch foundation 'Family and patient Centred Intensive Care' (FCIC; one representative is a formal representative for FCIC); (3) act as members of the Stakeholders group to provide advice on the study contents, execution and course on a regular basis to ensure the patient and family perspective; (4) advise on the contents of the patient information form and the informed consent procedure; (5) advise on ways to minimise loss to follow-up for the functional and cognitive outcome assessments; (6) advise on contents and organisation of symposia during the study on delirium and its consequences with the aim to better inform participants of the study and their family members and maximise their involvement; and (7) advise on the contents of the supporting website of the trial. Study participants will be informed about the most important results of the trial, either by post or symposium, when they indicate this on the informed consent letter.

## ETHICS AND DISSEMINATION

The study has been approved by the Medical Ethics Committee of the Erasmus University Medical Centre Rotterdam (MEC2017-511) and the Institutional Review Boards of participating sites. The study will be conducted according to the principles of the Declaration of Helsinki (version, date, see for the most recent version: www.wma.net) and in accordance with the Medical Research Involving Human Subjects Act (WMO) and other guidelines, regulations and Acts.

### Recruitment and consent

Recruitment of eligible patients will be done at admission. Informed consent for *possible* participation (ie, only when participants develop delirium at the ICU) will be obtained from subjects who are not expected to leave the

ICU within the first 24 hours after admission and are not yet delirious. Informed consent will be obtained from the patient or (if the patient is unable to consent) from the patient's representative. This procedure of prior request for informed consent will facilitate randomisation when the patient indeed develops delirium because randomisation can then be performed 24/7 since informed consent is already obtained and delirium often surfaces during the evening and night when obtaining informed consent is difficult. The informed consent procedure will be clearly delineated from the randomisation procedure. Importantly, when a patient with prior informed consent develops delirium and can thus be randomised, still a pre-randomisation check with regard to inclusion and exclusion criteria will be performed to confirm that the patient fulfils the inclusion, and not the exclusion criteria (because this may change over time). A team of dedicated research and ICU nurses and physicians (local principal investigator (PI), PhD student, PI, post-doc) will be trained to perform the informed consent procedures and help with the randomisations. Moreover, a 24/7 study consultation telephone number will be opened to help with problems or question during the study. A second type of randomisation concerns patients who are delirious at admission to ICU. These patients' next-of-kin will be asked to grant permission to participate by means of informed consent when they are legally representative for the patients and the patient has no contraindications. After informed consent is obtained, the patient can be randomised.

## Author affiliations
[1]Department of Intensive Care Adults, Erasmus MC- University Medical Center, Rotterdam, Zuid-Holland, Netherlands
[2]Department of Pharmacy and Health Systems Sciences, Northeastern University Bouve College of Health Sciences, Boston, Massachusetts, USA
[3]Division of Pulmonary, Critical Care and Sleep Medicine, Tufts Medical Center, Boston, Massachusetts, USA
[4]Department of Psychiatry, Erasmus MC - University Medical Center, Rotterdam, Zuid-Holland, Netherlands
[5]Department of Intensive Care, Albert Schweitzer Hospital Location Dordwijk, Dordrecht, Zuid-Holland, Netherlands
[6]Department of Intensive Care Medicine and UMC Utrecht Brain Center, University Medical Centre Utrecht Brain Centre, Utrecht, Utrecht, Netherlands
[7]Department of Pharmacy and Department of Intensive Care Adults, Erasmus MC - University Medical Center, Rotterdam, Zuid-Holland, Netherlands

**Collaborators** EuRIDICE study group authors: Local principal investigators: M. van den Boogaard (Radboud University Medical Center, Department of Intensive Care Medicine, Nijmegen, The Netherlands); A.J.B.W. Brouwers (Franciscus Gasthuis, Department of Intensive Care, Rotterdam, Netherlands); J.A. Lens (IJsselland Hospital, Department of Intensive Care, Capelle aan den IJssel, Netherlands); B.J.M. van der Meer (Maasstad Hospital, Department of Intensive Care, Rotterdam, Netherlands) H. Ponssen (Albert Schweitzer Hospital, Department of Intensive Care, Dordrecht, Netherlands); F.J. Schoonderbeek (Ikazia Hospital, Department of Intensive Care, Rotterdam, Netherlands); K.S. Simons (Jeroen Bosch Hospital, Department of Intensive Care Medicine, 's-Hertogenbosch, The Netherlands). Local research nurses: E. Berger (Franciscus Gasthuis, Department of Intensive Care, Rotterdam, Netherlands); A. Bouman (Jeroen Bosch Hospital, Department of Intensive Care Medicine, 's-Hertogenbosch, The Netherlands); M. Campo (Maasstad Hospital, Department of Intensive Care, Rotterdam, Netherlands); D. van Duijn (Erasmus MC-University Medical Center, Department of Intensive Care Adults, Rotterdam, The Netherlands, Netherlands); H. Embden – van Donk (Ikazia Hospital, Department of Intensive Care, Rotterdam, Netherlands); D. van de Graaf (IJsselland Hospital, Department of Intensive Care, Capelle aan den IJssel, Netherlands); E. Hoogendoorn (Albert Schweitzer Hospital, Department of Intensive Care, Dordrecht, Netherlands); P. Ormskerk (Erasmus MC-University Medical Center, Department of Intensive Care Adults, Rotterdam, The Netherlands, Netherlands); N. Roovers (Radboud University Medical Center, Department of Intensive Care Medicine, Nijmegen, The Netherlands); E. Toscano (Maasstad Hospital, Department of Intensive Care, Rotterdam, Netherlands); A. Vileito (Erasmus MC-University Medical Center, Department of Intensive Care Adults, Rotterdam, The Netherlands, Netherlands); T. van Zuylen (Jeroen Bosch Hospital, Department of Intensive Care Medicine, 's-Hertogenbosch, The Netherlands). All involved local principal investigators (MB, AB, BM, JL, EK, FS, KS) and research nurses (EB, AB, DD, HE, DG, PO, NR, AV, MC, ET, EH, TZ) have facilitated visits at their site, will be involved in executing the protocol at their sites and will be responsible for patient recruitment and data collection in their hospitals, along with follow up of study patients. Other members involved: C. Exler (Erasmus MC-University Medical Center, Department of Pharmacy, Rotterdam, The Netherlands, Netherlands); E. van den Berg (Erasmus MC-University Medical Center, Department of Neuropsychology, Rotterdam, The Netherlands, Netherlands); J. van Meeteren (Erasmus MC-University Medical Center, Department of Rehabilitation Medicine, Rotterdam, The Netherlands, Netherlands); M. Koopmanschap (Erasmus School of Health Policy & Management, Department of Health Economics and HTA, Rotterdam, The Netherlands, Netherlands). Patient representatives: I. Nutma and E. Kuijper.

**Contributors** MvdJ developed the study protocol, edited and approved the final version, and was responsible for funding and supervising the study coordination. ZT contributed to the protocol development and study implementation. LS is responsible for study coordination. WJRR assisted with statistical analysis and NGMH assisted with pharmacological coordination. R-JO, AJCS, HHP and JWD were involved in the study design and protocol development. All authors (LS, ZT, JWD, R-JO, HHP, AJCS, NGMH, WJRR, DG, MvdJ) contributed to the development and refinement of this study protocol. They have read and approved the final version of the protocol.

**Funding** A grant has been provided by ZonMw – The Netherlands Organisation for Health Research and Organisation. ZonMw project number: 848041001.

**Disclaimer** The funding source had no role in the design of this study and will not have any role during its execution, analyses, interpretation of the data or decision to submit results.

**Competing interests** None declared.

**Patient and public involvement** Patients and/or the public were involved in the design, or conduct, or reporting, or dissemination plans of this research. Refer to the Methods section for further details.

**Patient consent for publication** Not required.

**Provenance and peer review** Not commissioned; externally peer reviewed.

**ORCID iDs**
Lisa Smit http://orcid.org/0000-0001-5199-930X
Wim J R Rietdijk http://orcid.org/0000-0002-2622-7321

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
