## [Reviewer comments · BMJ Open]

ARTICLE DETAILS

TITLE (PROVISIONAL)	Efficacy of haloperidol to decrease the burden of Delirium In adult Critically ill patiEnts (EuRIDICE): study protocol for a prospective randomised multi-center double-blind placebo-controlled clinical trial in the Netherlands
AUTHORS	Smit, Lisa; Trogrlić, Zoran; Devlin, John; Osse, Robert-Jan; Ponsen, Huibert; Slooter, Arjen; Hunfeld, Nicole; Rietdijk, Wim; Gommers, Diederik; van der Jagt, Mathieu

VERSION 1 – REVIEW

REVIEWER	Mark Oldham University of Rochester Medical Center, USA
REVIEW RETURNED	02-Apr-2020

GENERAL COMMENTS	The authors present the protocol from the ongoing EuRIDICE study, which is a randomized, multi-center, double-blind, placebo-controlled trial of haloperidol for ICU delirium, being conducted throughout the Netherlands. Subjects are enrolled at the time of ICU admission for possible inclusion but randomized to begin receiving scheduled study drug (either intravenous haloperidol or placebo for up to 14 days) when they are first found to have delirium. For those with delirium at the time of admission (prevalent delirium), randomization occurs on ICU admission; otherwise, subjects who develop delirium in the ICU (incident delirium) are randomized to begin receiving a study drug when delirium is first detected. The primary outcome of this study is delirium- and coma-free days in the ICU. Many aspects of this current study parallel the recent MIND-USA study (Girard NEJM 2018) and, specifically with regard to the assessment of post-ICU mortality, the REDUCE study (van den Boogaard JAMA 2018). Notably, though, in EuRIDICE and MIND-USA, haloperidol is given to patients with delirium as treatment; in REDUCE, haloperidol was given prophylactically to patients at high risk for delirium. It also bears noting that EuRIDICE is, in many ways, analogous to the ongoing multi-national European AID-ICU trial (http://www.cric.nu/aid-icu-protocol/). As this is an ongoing study, my comments below are aimed only at ensuring clarity of the protocol for readers. 1) Under exclusion criteria at the time of randomization, please define “(recent)” in relation to torsade de pointes (#2) and neuroleptic malignant syndrome or parkinsonism (#3). Additionally, in the US at least, “neuroleptic malignant syndrome” is the standard ordering of these words. 2) Please review the use of the term “endpoints” (P8, LL163, 164, 167) and whether “outcomes” may be preferred in this context.
---

	3) The “Secondary study endpoints” section appears to include both baseline values (e.g., demographics and ICU days before study entry, RASS [including at randomization]) as well as study outcomes of interest (e.g., subsequent RASS scores, use of “escape medication,” daily respiratory status). Consider separating these so it’s clear which are to be reported to characterize the sample, perhaps which will be included as a priori covariates in multivariate models, and which are secondary outcomes. 4) The following pertain to “Dosages, dosage modifications and method of administrations” (p11, LL239–247) section: a. The section indicates the study drug will be increased “when delirium persists beyond the next 8-hour shift.” Please clarify that this is equivalent to saying “if the subsequent delirium assessment is positive” or whether “beyond the next 8-hour shift” is intended to mean “if any subsequent delirium assessment is positive,” including even after one or two consecutive negative delirium assessments (as these would not prompt study drug discontinuation). b. Similarly, please clarify that the discontinuation parameter of “when delirium has resolved for the next 24 hours” is equivalent to “after three consecutive negative delirium assessments.” c. Please comment on how coma is being handled for subsequent dose modifications. 5) Under “General medical management at participating ICUs” (P13, LL288-289), the authors indicate that they will be performing “spot-checks” to “confirm delirium screening accuracy.” I would ask if the authors could specify the frequency of these assessments or what proportion of overall assessments will be assessed? Similarly, is this being performed to assess inter-rater reliability in a formal manner? 6) I appreciate the detailed “escape medication” section. I would note that quetiapine is available only in oral formulation, and the authors may wish to clarify what instruction is given to providers where a parenteral option is indicated in the treatment team’s estimation. Would parenteral haloperidol be the preferred agent in such instances even though “strongly discouraged” (L315)? 7) Under Statistical Analysis (p18, LL425-437), the authors indicate that “A more detailed statistical plan, to be drawn up before Data Base Lock, will be drafted for publication separately.” I’m curious if at this stage the authors could identify which covariates they plan to assess for inclusion in Poisson regression and what their criteria for inclusion will be? Minor textual points: P8, L156: please change “alfa” to “alpha” P9, L172: the second “S” in RASS is “Scale” P9, L181: “self-extubation” would be more familiar to an English-speaking audience than “auto-extubation” P10, L210: “Revisited” should be “Revised” I would characterize the neuropsychological battery being performed as “detailed,” though I note that the authors describe it as “extensive” (under study strengths, p5, L68) but elsewhere “brief” (under outcomes, p10, LL199 & 212).
--	---

REVIEWER	Martin Siegemund Intensive Care Unit University Hospital Basel Department of Clinical Research University of Basel Switzerland
REVIEW RETURNED	04-Apr-2020

GENERAL COMMENTS

The manuscript «Efficacy of haloperidol to decrease the burden of Delirium

In adult Critically ill patients (EuRIDICE): study protocol for a prospective randomised multi-center double-blind placebo-controlled clinical trial” is the protocol of an already started study. This study wants to investigate the possible therapeutic role of haloperidol in intensive care patients with delirium. This is a truly valuable study, because haloperidol is used for the treatment of delirium in many intensive care units all over the world, despite a lack of data from well conducted, prospective randomized trials. The trials used haloperidol in cohorts of not appropriate (already sedated) patients or focused on the prophylaxis of delirium with haloperidol. From a theoretical, pharmacologic point of view haloperidol should be a useful drug to treat patients at least with a hyperactive or mixed phenotype of delirium. Moreover, the success of haloperidol for this indication seems to be evident in many intensive care units, otherwise the common use of the drug in the light of the scarce and negative evidence is inexplicable. The current manuscript describes a double blind, placebo controlled trial to elucidate the capabilities of haloperidol in the treatment of intensive care associated delirium.

The study protocol of the EuRIDICE study is well written and the study will give urgently required evidence about the use of haloperidol for intensive care associated delirium. The planned sample size and the exclusion criteria seem adequate to find a possible existing effect of haloperidol in the treatment of delirium. The statistical plan to analyse the data with a Poisson regression is adequate to analyse counted data like days free of delirium. A clear strength of the study is the large amount of neuro-psychological and cognitive tests the authors plan to administer after 3 and 12 month to elucidate long-term sequelae of delirium.

There are a few points that should be considered; although the trial has already started and principal modifications of the protocol are no longer possible.

1. The authors do not discuss if there will a different treatment of patients with hypoactive delirium. In a patient already secluded and inactive, the effect of haloperidol may not be as obvious as in patients with an agitated or mixed form of delirium. The authors should clearly pre-plan a subgroup analysis of patients with hypoactive delirium compared to the other two phenotypes of delirium.
2. The authors state that the study statistician has access to the data to unblind a patient in the case of a SUSAR or SAE. This seems not to be permissible. The statistician should be blinded to the group affiliation of all randomized patients until the end of the study. Study analysis should also be carried out in a blinded manner, and the authors should write a blinded abstract for attention of the data safety monitoring board before breaking the randomization code.
3. The authors want to assess “ICU delirium and coma free days up to 14 days after randomisation”. This seems appropriate to compensate competing risks like death. The authors do not describe how they do assess delirium free days in patients discharged before day 14 and how they control for patients discharged to the ward still with delirium.
4. Abstract: «Main study endpoint is ICU days free of delirium and coma (DCFD)” should be changed in “Main study endpoint is delirium and coma free days (DCFD) in 14 days after study inclusion.” to clear the abbreviation.

	5. SAE: If pregnancy and lactation is an exclusion criterion, why are congenital anomaly and birth defect listed in the SAE section? 6. Appendix 4 seems to be copied from a grant application (5. Added value) and should be adapted for publication in BMJ OPEN 7. Appendix 5: It is not clearly described how the study drug is prepared. Does the pharmacy provide boxes with 42 doses (3x 14) of 5 mg ampules of haloperidol? Moreover, how is ensured that the patient receives the drug out of the box attributed to him?
--	---

VERSION 1 – AUTHOR RESPONSE

Reviewer 1

Under exclusion criteria at the time of randomization, please define “(recent)” in relation to torsade de pointes (#2) and neuroleptic malignant syndrome or parkinsonism (#3). Additionally, in the US at least, “neuroleptic malignant syndrome” is the standard ordering of these words.

Response

We have changed the order of the words of “malignant neuroleptic syndrome”.

With regards to the definition of “recent” for randomisation exclusion criteria #2 and #3, we would like to point out that these two criteria are also exclusion criteria for eligibility. We included these two criteria as exclusion criteria for randomisation, as torsade de pointes, neuroleptic malignant syndrome and parkinsonism may occur during ICU stay after patients have been screened for eligibility (which is as soon as possible after ICU admission). To prevent patients who experienced one of these criteria from being randomised, we included these two criteria as exclusion criteria for randomisation. Taking this into account, “recent” is considered as at any point after screening for eligibility and up to the moment of randomisation, which is a necessary consequence of application of the eligibility criteria first and the randomisation criteria subsequently.

Reviewer1, cont’d

Please review the use of the term “endpoints” (P8, LL163, 164, 167) and whether “outcomes” may be preferred in this context.

Response

Thank you for this suggestion. We have replaced the term “endpoints” with “outcomes”:

“Study outcomes

Main study outcome:

[...]

Secondary study outcomes:”

Reviewer1, cont’d

The “Secondary study endpoints” section appears to include both baseline values (e.g., demographics and ICU days before study entry, RASS [including at randomization]) as well as study outcomes of interest (e.g., subsequent RASS scores, use of “escape medication,” daily respiratory status). Consider separating these so it’s clear which are to be reported to characterize the sample, perhaps which will be included as a priori covariates in multivariate models, and which are secondary outcomes.

Response

We have adjusted our secondary study outcome paragraph according to this comment and moved the baseline demographics to the “statistical analysis” section in the protocol.

Reviewer1, cont'd

The following pertain to “Dosages, dosage modifications and method of administrations” (p11, LL239–247) section:

- a. The section indicates the study drug will be increased “when delirium persists beyond the next 8-hour shift.” Please clarify that this is equivalent to saying “if the subsequent delirium assessment is positive” or whether “beyond the next 8-hour shift” is intended to mean “if any subsequent delirium assessment is positive,” including even after one or two consecutive negative delirium assessments (as these would not prompt study drug discontinuation).
- b. Similarly, please clarify that the discontinuation parameter of “when delirium has resolved for the next 24 hours” is equivalent to “after three consecutive negative delirium assessments.”
- c. Please comment on how coma is being handled for subsequent dose modifications.

Response

We acknowledge the point raised by the reviewer regarding the phrasing here, therefore we have changed it to clarify better how the study dosing was conducted:

“ The following dosing scheme will be used: start with haloperidol/placebo (further called: “study drug”) 2.5mg IV q8h (because of delirium screening once every 8-hour shift) and increase to a maximum dose of 5mg IV q8h when delirium persists during the next 8-hour shift. Doses will be reduced (50% of dose) in the very old elderly (age \geq 80 years). The study drug dose will be decreased (when dosage is 5mg IV q8h) or stopped (when dose is 2.5mg IV q8h) when delirium has resolved (or is un-assessable due to coma) for the next 24 hours (implying: three consecutive delirium assessments during three shifts).”

Reviewer1, cont'd

Under “General medical management at participating ICUs” (P13, LL288-289), the authors indicate that they will be performing “spot-checks” to “confirm delirium screening accuracy.” I would ask if the authors could specify the frequency of these assessments or what proportion of overall assessments will be assessed? Similarly, is this being performed to assess inter-rater reliability in a formal manner?

Response

These spot-checks are conducted once or twice during the trial. The primary aim of these spot-checks is to encourage ICU nurses to be aware of the quality of delirium assessments, to promote discussion about any arisen difficulties in daily practice and to create the opportunity to ask questions, as a quality assessment given the fact that it concerns the primary outcome:

“During the inclusion period of the current trial, spot-checks will be performed by members of the investigative team at each center to confirm delirium screening accuracy, as a quality-of-assessments measure and these will be documented in a qualitative manner.”

Reviewer1, cont'd

I appreciate the detailed “escape medication” section. I would note that quetiapine is available only in oral formulation, and the authors may wish to clarify what instruction is given to providers where a parenteral option is indicated in the treatment team’s estimation. Would parenteral haloperidol be the preferred agent in such instances even though “strongly discouraged” (L315)?

Response

Within the boundaries of the protocol, and when the maximum dosage of 5mg IV q8h of study drugs would have been reached, persistent agitation or hallucinations would be managed at the discretion of the treating physician, whilst still avoiding haloperidol. Alternatives for agitation would include IV drips of clonidine, dexmedetomidine, propofol or midazolam, or combining/increasing medications for sleep. Management of persistent hallucinations would include finding a better way to administer oral drugs, e.g. by inserting a deep feeding tube beyond the pyloric sphincter (duodenal tube). Otherwise we

would consult with the attending psychiatrist for alternatives, e.g. olanzapine. However, in our experience refractory hallucinations did not seem to be a very frequent issue in the patients included.

Reviewer1, cont'd

Under Statistical Analysis (p18, LL425-437), the authors indicate that "A more detailed statistical plan, to be drawn up before Data Base Lock, will be drafted for publication separately." I'm curious if at this stage the authors could identify which covariates they plan to assess for inclusion in Poisson regression and what their criteria for inclusion will be?

Response

We will adjust for any significant differences in baseline characteristics between the haloperidol and placebo group, based on the baseline demographics described in the protocol. We will consult with our statistician before database lock how to do the analysis. We will include known prognostic variables of disease severity (age, APACHE IV, admission diagnosis category). However, we hope adjustment will not be a necessity due to the fact that this is a randomized study and prognostic variables at baseline should be balanced.

Reviewer1, cont'd

Minor textual points:

P8, L156: please change "alfa" to "alpha"

P9, L172: the second "S" in RASS is "Scale"

P9, L181: "self-extubation" would be more familiar to an English-speaking audience than "auto-extubation"

P10, L210: "Revisited" should be "Revised"

I would characterize the neuropsychological battery being performed as "detailed," though I note that the authors describe it as "extensive" (under study strengths, p5, L68) but elsewhere "brief" (under outcomes, p10, LL199 & 212).

Response

We like to thank the reviewer for is detailed assessment of the manuscript. We have processed these textual points in the manuscript.

Reviewer 2

The authors do not discuss if there will a different treatment of patients with hypoactive delirium. In a patient already secluded and inactive, the effect of haloperidol may not be as obvious as in patients with an agitated or mixed form of delirium. The authors should clearly pre-plan a subgroup analysis of patients with hypoactive delirium compared to the other two phenotypes of delirium.

Response

Indeed, all delirium subtypes are randomized in the EuRIDICE trial, meaning that patients with hypoactive delirium, as well as patients with hyperactive and mixed delirium, will all receive the same treatment (either haloperidol or placebo). We have indeed planned a sensitivity analysis to investigate whether there is a difference between the three subtypes, and some other subgroups, with regard to efficacy of haloperidol. We thank the reviewer for pointing out that this was not yet included in the manuscript. We have now added this to the Statistical Analysis paragraph:

"Pre-defined sub-analyses will include efficacy stratified by 1) agitated, mixed-type or hypoactive delirium; 2) the presence of hallucinations or delusions; 3) delirium severity (based on ICDSC score); and 4) sedation-related, hypoxic, metabolic or septic delirium."

Reviewer 2, cont'd

The authors state that the study statistician has access to the data to unblind a patient in the case of a SUSAR or SAE. This seems not to be permissible. The statistician should be blinded to the group

affiliation of all randomized patients until the end of the study. Study analysis should also be carried out in a blinded manner, and the authors should write a blinded abstract for attention of the data safety monitoring board before breaking the randomization code.

Response

We thank the reviewer for his astute assessment. The study statistician was involved in generating the randomisation list and hence is aware of the contents of each medication kit. However, the trial statistician is not involved with study procedures or unblinding in case of SUSAR or SAEs, this has now been adjusted in the manuscript. The randomization code will be broken at time of Database Lock, at which point all data for analysis will have been collected. Afterwards, study analysis will be performed in an unblinded manner, as we will not be able to analyse study results if we remain unaware of the treatment groups.

“Only the involved pharmacists and the trial statistician are aware of the contents of each medication kit. Only the local (site) pharmacists were able to unblind study treatment of a patient in case of an emergency.”

Reviewer 2, cont'd

The authors want to assess “ICU delirium and coma free days up to 14 days after randomisation”. This seems appropriate to compensate competing risks like death. The authors do not describe how they do assess delirium free days in patients discharged before day 14 and how they control for patients discharged to the ward still with delirium.

Response

Patients who are discharged before the 14 day study period has ended, will be recorded as delirium and coma-free after discharge. Additionally, we will assume all patients who died within 14 days after randomisation to have 0 delirium and coma free days, which is comparable to similar studies (Girard et al., Crit Care Med 2010; Page et al., Lancet Respir Med 2013; van den Boogaard et al., JAMA 2018). We have added this to our “Statistical analysis” section.

“Patients who are discharged before the 14 day study period has ended, will be recorded as delirium and coma-free after discharge (8, 35). Additionally, we will assume all patients who died within 14 days after randomisation to have 0 delirium and coma free days (7).”

Reviewer 2, cont'd

Abstract: «Main study endpoint is ICU days free of delirium and coma (DCFD)” should be changed in “Main study endpoint is delirium and coma free days (DCFD)in 14 days after study inclusion.” to clear the abbreviation.

Response

We adapted the abstract with the suggestion:

“Main study endpoint is delirium and coma free days (DCFD) up to 14 days after randomisation.”

Reviewer 2, cont'd

SAE: If pregnancy and lactation is an exclusion criterion, why are congenital anomaly and birth defect listed in the SAE section?

Response

Congenital anomaly or birth defect is part of the definition of a SAE. However, indeed, this SAE is not applicable for the EuRIDICE trial. We have specified the SAEs for the purpose of the study in Italics in this section of the protocol. For the specific SAE “congenital anomaly or birth defect” we have added that this SAE is not applicable (N/A):

“is a congenital anomaly or birth defect; (Not applicable) or”

Reviewer 2, cont'd

Appendix 4 seems to be copied from a grant application (5. Added value) and should be adapted for publication in BMJ OPEN

Response

We have adapted Appendix 4 by leaving out section "5. Added value". The other sections of Appendix 4 remained unaffected in order to provide the reader with available literature up to the date that the protocol was developed.

Reviewer 2, cont'd

Appendix 5: It is not clearly described how the study drug is prepared. Does the pharmacy provide boxes with 42 doses (3x 14) of 5 mg ampules of haloperidol? Moreover, how is ensured that the patient receives the drug out of the box attributed to him?

Response

We have specified these procedures in the paragraph "Randomisation, blinding and treatment allocation", and Appendix 5:

"Upon randomisation, the study drug with the corresponding randomisation kit number 1-8 (based on 8 medication batches consisting of either haloperidol or placebo) will be obtained from the hospital pharmacy of each participating ICU. Each box from a batch/kit contains 10 ampules (5mg/1ml) of haloperidol or placebo. If all ampules are used, a new box from the same medication kit number with 10 ampules will be used. Study drugs are administered on prescription in the electronic patient data management system (PDMS) and are double-checked by ICU nurses before administration, which is similar to regular practice. Furthermore, the kit number was noted upon randomisation in the medical file and the kit number could be retrieved at any time from the PDMS after first prescription upon randomisation."

We hope the revisions are sufficient and have improved the manuscript with the help of the useful reviewers' comments and that the revised manuscript will be considered suitable for publication in BMJ Open.

VERSION 2 – REVIEW

REVIEWER	Mark Oldham University of Rochester Medical Center, USA
REVIEW RETURNED	20-Apr-2020

GENERAL COMMENTS	I appreciate the authors' revisions for clarification. I have no further substantive comments. I offer the authors one very minor suggestion: In the "strengths and limitations of this study" section, it would read better if the word "European" in the first bullet is removed and added as "in Europe" at the end of the clause: "This is the first sufficiently powered...trial in Europe."
--

REVIEWER	Prof Dr Martin Siegemund Department of Intensive Care University Hospital Basel Department of Clinical Research University of Basel Basel Switzerland
-----------------	---

	Sponsor of two randomized, controlled trials about medical treatment of either hypoactive or hyperactive and mixed delirium using melatonin or dexmedetomidine and propofol respectively. No further commercial or scientific conflict of interest
REVIEW RETURNED	06-May-2020

GENERAL COMMENTS	The authors adequately addressed all suggestions of the reviewers, so that the manuscript is now nearly ready for publication. In their statistical analysis paragraph they newly state about their sensitivity analysis, as recommended: "Pre-defined sub-analyses will include efficacy stratified by 1) agitated, mixed-type or hypoactive delirium; 2) the presence of hallucinations or delusions; 3) delirium severity (based on ICDSC score); and 4) sedation-related, hypoxic, metabolic or septic delirium." If the authors really plan a sub-group analysis stratified by ICDSC score based delirium severity, they should affiliate a sentence how they classify delirium severity, because ICDSC scores of 4-5 could be compiled in patients with mixed or hyperactive as well as hypoactive delirium.
--

VERSION 2 – AUTHOR RESPONSE

Reviewer 1

I offer the authors one very minor suggestion: In the "strengths and limitations of this study" section, it would read better if the word "European" in the first bullet is removed and added as "in Europe" at the end of the clause: "This is the first sufficiently powered...trial in Europe."

Response

Thank you for your previous comments to improve our manuscript, and thank you for this suggestion. We have changed the mentioned sentence in the "Strengths and limitations" section: "This study is the first European sufficiently powered randomised multi-center double-blind placebo-controlled clinical trial **in Europe**."

Reviewer 2

In their statistical analysis paragraph they newly state about their sensitivity analysis, as recommended: "Pre-defined sub-analyses will include efficacy stratified by 1) agitated, mixed-type or hypoactive delirium; 2) the presence of hallucinations or delusions; 3) delirium severity (based on ICDSC score); and 4) sedation-related, hypoxic, metabolic or septic delirium." If the authors really plan a sub-group analysis stratified by ICDSC score based delirium severity, they should affiliate a sentence how they classify delirium severity, because ICDSC scores of 4-5 could be compiled in patients with mixed or hyperactive as well as hypoactive delirium.

Response

We would like to thank you for your current and previous suggestions. With regards to delirium severity, we added definitions regarding delirium severity. For our study we will use the ICDSC to indicate the level of delirium severity in ICU patients (Gusmao-Flores 2013 Rev Bras Ter Intensiva). Delirium severity will only be studied in those patients screened with ICDSC and not in those screened with the CAM-ICU, since the latter can only be scored either as positive or negative and has no nominal value to indicate severity, in contrast to ICDSC. Further, we will separately analyse the subtypes of delirium separately regarding effects of haloperidol in a sensitivity analyses.

"Pre-defined sub-analyses will include efficacy stratified by 1) agitated, mixed-type or hypoactive delirium; 2) the presence of hallucinations or delusions; 3) delirium severity (based on ICDSC score: **low delirium severity = mean ICDSC score of 4 to 5; medium delirium severity = mean ICDSC score 5 to 7; or high delirium severity = ICDSC score 7 to 8**); and 4) sedation-related, hypoxic, metabolic or septic delirium."

1